# Current Knowledge on the Oxidative-Stress-Mediated Antimicrobial Properties of Metal-Based Nanoparticles

**DOI:** 10.3390/microorganisms10020437

**Published:** 2022-02-14

**Authors:** Nour Mammari, Emmanuel Lamouroux, Ariane Boudier, Raphaël E. Duval

**Affiliations:** 1Université de Lorraine, CNRS, L2CM, F-54000 Nancy, France; nour.mammari@univ-lorraine.fr (N.M.); emmanuel.lamouroux@univ-lorraine.fr (E.L.); 2Université de Lorraine, CITHEFOR, F-54000 Nancy, France; ariane.boudier@univ-lorraine.fr; 3ABC Platform®, F-54505 Vandœuvre-lès-Nancy, France

**Keywords:** metal-based nanoparticles, oxidative stress, ROS, antibacterial activity, antibacterial mechanisms

## Abstract

The emergence of multidrug-resistant (MDR) bacteria in recent years has been alarming and represents a major public health problem. The development of effective antimicrobial agents remains a key challenge. Nanotechnologies have provided opportunities for the use of nanomaterials as components in the development of antibacterial agents. Indeed, metal-based nanoparticles (NPs) show an effective role in targeting and killing bacteria via different mechanisms, such as attraction to the bacterial surface, destabilization of the bacterial cell wall and membrane, and the induction of a toxic mechanism mediated by a burst of oxidative stress (e.g., the production of reactive oxygen species (ROS)). Considering the lack of new antimicrobial drugs with novel mechanisms of action, the induction of oxidative stress represents a valuable and powerful antimicrobial strategy to fight MDR bacteria. Consequently, it is of particular interest to determine and precisely characterize whether NPs are able to induce oxidative stress in such bacteria. This highlights the particular interest that NPs represent for the development of future antibacterial drugs. Therefore, this review aims to provide an update on the latest advances in research focusing on the study and characterization of the induction of oxidative-stress-mediated antimicrobial mechanisms by metal-based NPs.

## 1. Introduction

The World Health Organization (WHO) states that the indiscriminate use of antibiotics has facilitated increasing bacterial resistance, and this has become a serious public health problem worldwide. Microbes, mainly bacteria, are now frequently resistant to several antibiotics; consequently, therapeutic options become more and more limited, and, concomitantly, nosocomial infections more and more severe [1]. The situation is so critical that in 2017 WHO published a list of 12 resistant bacteria which represent a real threat to human health [1,2]. These bacteria are divided by WHO into three groups according to their emergency profile in relation to resistance to antibiotics. The first group is bacteria classified as a priority, also called “Critical Urgency” and includes: *Acinetobacter baumannii*, *Pseudomonas aeruginosa* and *Enterobacteriaceae*. They are Gram-negative bacteria, carbapenem-resistant, and third-generation-cephalosporin resistant. The second group is “High Urgency” and includes: vancomycin-resistant *Enterococcus faecium*, methicillin- and vancomycin-resistant *Staphylococcus aureus*, clarithromycin-resistant *Helicobacter pylori*, fluoroquinolone-resistant *Campylobacter* spp. and *Salmonella* spp., and third-generation-cephalosporin- and fluoroquinolone-resistant *Neisseria gonorrhoeae*. The third group is “Medium Urgency” and includes: penicillin-non-susceptible *Streptococcus pneumoniae*, ampicillin-resistant *Haemophilus influenzae*, and fluoroquinolone-resistant *Shigella* spp. [1,2].

Moreover, researchers have indicated that bacterial resistance to antibiotics due to genetic modification is correlated mainly with antibiotic consumption, and abundant antibiotic prescriptions are associated with the development of antibiotic resistance [3]. It is also important to note that the approval of new antibacterial agents with new mechanisms of action by the Food and Drug Administration (FDA) has declined since 1983 (even though we have observed a slight rebound in recent years) [2]. Hence, it is necessary to initiate preventive actions against MDR bacteria and to develop new molecules effective against those pathogens in order to reduce the deathrate of people.

In this context, nanoparticles (NPs) that have been shown to effectively target and destroy microbes represent a promising solution [4]. Indeed, metal-based NPs, such as metal-oxide nanoparticles (MONPs), and biosynthesized NPs and their recombinants have been used to overcome limitations of classic antibacterial drugs [5]. Furthermore, studies on metal-based NPs, including those with gold (Au), silver (Ag), iron (Fe), copper (Cu), magnesium (Mg), and MONPs with zinc oxide (ZnO), iron oxide (e.g., Fe_3_O_4_ or α-Fe_2_O_3_), titanium dioxide (TiO_2_), and cerium oxide (CeO_2_), indicate their potential use as novel antimicrobial agents for the control of microorganisms [6]. Metal-based NPs have shown an ability to inhibit bacterial growth and bacterial biofilm formation [7,8,9,10,11,12] and consequently have been proposed as tools to combat infectious diseases [13].

Thus, many studies have abundantly demonstrated the antimicrobial activity of metal-based NPs [4,6,14]. The antimicrobial mechanism of action of metal-based NPs is described as being mainly linked to the following sequence of reactions: attraction to the bacterial surface, destabilization of the bacterial cell wall and membrane, resulting in a change in its permeability, induction of toxicity and oxidative stress by generation of reactive oxygen species (ROS) and free radicals, and, finally, the modulation of signal transduction pathways (Figure 1) [15]. Numerous works also show that NP-based materials have been used as a new defense strategy against antimicrobial resistance.

NPs have been described to involve different mechanisms for combating microbial resistance, such as oxidative stress [5]. Major involvement of ROS in antimicrobial therapy could be an interesting antibacterial strategy; therefore, it is essential to fully demonstrate that the metal-based NPs could kill the bacteria via a mechanism of action dependent on the oxidative stress. The main objective of this review is to provide advances focused on antibacterial mechanism of action related to ROS production by metal-based nanoparticle treatment. The secondary objective is to specify the method used to assess ROS production in bacteria following metal-based NP treatment.

## 2. Mechanistic Insights into the Antimicrobial Actions of Metal-Based Nanoparticles and Oxidative Stress Due to ROS Generation

### 2.1. Antibacterial Mechanisms of NPs

Most antimicrobial drugs target bacteria by inhibition of (i) cell wall synthesis, with peptidoglycan as the main target [16,17,18], (ii) nucleic acid synthesis, (iii) protein synthesis [19], and (iv) modification of membrane permeability [20]. Bacteria are particularly adept at developing or acquiring resistance mechanisms independent of target functionalities. The various resistance mechanisms included: expression of enzymes capable of altering or degrading antimicrobial agents [21,22], causing cell wall modifications, ribosomal mutations [23], and changes in porin expression (with or without an active efflux mechanism) [24,25].

The use of metal-based NPs as antibacterial agents opened opportunities to develop new strategies against antimicrobial resistance [14]. Even if their antibacterial mechanisms of action are not well elucidated, the majority of the evidence shows that the NPs act when they are in contact with the bacterial cell walls via various antimicrobial means [26]. It has been described that the negatively charged molecules composing both Gram-positive and Gram-negative cell walls promote the interaction between NPs and the bacterial membrane. These negatively charged molecules have a strong affinity for the positive ions released by most NPs, including AgNPs [27], AuNPs [28], CuNPs [6,29], ZnONPs [30,31], α-Fe_2_O_3_ metal-based NPs [32], and TiO_2_ [33], leading to the electrostatic attraction of NPs to the bacterial surface that induces a disruption of the bacterial cell wall and an increase in its permeability (Figure 1) [34]. It was determined that positively charged AgNPs are strongly attracted to the surface of the bacteria, which increases antibacterial activity [27]. Conversely, neutral or negatively charged AgNPs have significantly decreased antibacterial activity [27]. Once the contact between the NPs and the bacterial wall is established, the NPs can then directly cross the bacterial cell membranes, interfere with metabolic pathways, and induce changes in membrane shape and function. Once inside cells, NPs could inhibit enzymes, deactivate proteins, and induce oxidative stress and electrolyte imbalance [35]. NPs can also release metal ions in the extracellular space; these are capable of entering the cell and disrupting biological processes [34]. For example, it has been shown that during the treatment of *Escherichia coli* K12 with AgNPs, NPs interacted with the bacterial cell wall, then dissolved to release Ag^+^ into the cell, and finally triggered a transcriptional response that caused more toxicity to the cells [36,37]. It was demonstrated that bacterial DNA was condensed when both *E. coli* and *S. aureus* were exposed to Ag^+^, arresting cell multiplication [38]. In addition, there is evidence that exposing bacteria to NPs causes nuclear fragmentation [39] or physical attachment of the AgNPs to the DNA because of the high affinity of Ag^+^ to phosphates, which are highly abundant in DNA molecules [40]. Another study reported that *E. coli* treated with AgNPs upregulated many genes covering a wide range of cellular functions, including membrane structure, biofilm formation, the citric acid cycle, electron transfer, cellular transport, and protein efflux [41].

### 2.2. ROS-Dependent Oxidative Stress

“Oxidative stress” is described as an imbalance between the production and consumption of reactive oxygen species. ROS are chemically reactive species that include a variety of molecules and free radicals derived from molecular oxygen: hydrogen peroxide (H_2_O_2_), singlet oxygen (^1^O_2_), reactive superoxide radical anion (O_2_·^−^), and hydroxyl radicals (·OH) [42,43]. The reduction of molecular oxygen (O_2_) produces superoxide anion (O_2_·^−^), which is the precursor of most ROS and mediates chains of oxidation reactions. The dismutation of O_2_·^−^ leads to the formation of hydrogen peroxide (H_2_O_2_), which can be totally reduced to H_2_O or partially reduced to hydroxyl radicals (·OH). The formation of ·OH is also catalyzed by reducing transition metals (i.e., Fenton reaction). These transition metals can be reduced by the superoxide anion (O_2_·^−^), which inevitably leads to the propagation of the cascade of oxidation reactions (Figure 2) [44]. Alternatively, superoxide anion (O_2_·^−^) can react with nitric oxide radical (·NO) to form reactive nitrogen species, such as peroxynitrite anion (ONOO^−^), nitrogen oxide radical (·NO_2_^−^), nitrate anion (NO_3_^−^), and carbonate anion (CO_3_·^−^) [45]. This oxidative process can kill microorganisms if hydroxyl radical accumulation is not controlled and leads to oxidative damage of proteins, lipids, and nucleic acids [46]. Bacteria have a number of defense proteins against ROS, involving enzymes such as superoxide dismutase (SOD), catalase (CAT), glutathione peroxidase (GPx), glutathione reductase (GR) [47], the thioredoxin system [48], and peroxiredoxin (Prxs) [49]. When this system becomes ineffective for the detoxification of ROS, the bacterial cell oxidizes due to the stressful situation (Figure 1 and Figure 3).

### 2.3. Metal-Based Nanoparticles Target ROS Production in Bacteria

The generation of ROS has been described as an important mechanism of nanotoxicity, resulting in the subsequent induction of oxidative stress in cells and microbial death [50,51]. The level of ROS generation has been evaluated during treatment with nanomaterials, and it has been found to depend on the chemical nature of the NPs [51]. Engineered nanomaterial studies affirm that smaller size and high specific charge of the surface area leads to the production of higher levels of ROS [52,53]. Either metal ions or NPs can induce the production of ROS in the intracellular space, as described by Mazur et al. [54]. Another study on metal-based NPs showed that different NPs, while exhibiting similar inhibitory effects on bacterial growth, can differ significantly in terms of ROS-generated damage [55].

An increasing number of studies show that NP-induced oxidative stress can be exploited for killing a wide range of pathogens; thus, NPs could meet the need for new antibacterials with new mechanisms of action [56]. Several researchers report that AgNPs promote the induction of ROS [50,57]. The relevant antimicrobial effect of AgNPs is attributed to its potential upregulation of ROS, leading to damage of the cell cytoskeleton, proteins, and nucleic acids that eventually results in the antibacterial effect [50,58] (Figure 1). CuNPs are also well known to be highly cytotoxic by the release of ions and the production of ROS [59]. Additionally, other MONPs, such as ZnO [60], TiO_2_ [61], and MnO [62], have been proven to exhibit bactericidal activity via their ability to stimulate the cell to produce ROS (Figure 1). In addition to the disruption of bacterial membranes, NPs have been demonstrated to disrupt biofilm formation. In fact, biofilms play an important role in the development of bacterial resistance against antibiotics and other antibacterial agents [63]. Numerous studies have shown that metal-based NPs, including AuNPs [64], AgNPs [65], MgONPs [66], ZnONPs [67], and CuONPs [68], can prevent or overcome biofilm formation.

## 3. Overview of Recent Studies Evaluating ROS Production in Microbial Models

### 3.1. Method

In this context, a growing interest has been focused on the assessment of ROS production and activation of oxidative stress as an antibacterial mechanism following exposure to metal-based nanoparticles. In order to respond to our objectives, we performed a literature search (focusing on articles published between January 2018–November 2021) using the following databases: Medline, PubMed, and Scopus; we used the following keywords: “antibacterial activity”, “metallic nanoparticle”, and “ROS”. The exclusion of 128 duplicates resulted in 132 studies identified from a primary literature search. A total of 44 of the 132 records were excluded due to them being studies performed on human cells (*n* = 42) or reports not retrieved (*n* = 2). The remaining 88 studies were scanned, and 66 additional studies were excluded for the following reasons: not studying the antibacterial activity before measuring ROS, not measuring ROS, not detailing the ROS evaluation protocol, or not using metal-based nanoparticles. Finally, 22 studies were considered eligible for this literature review (Figure 3 and Table 1).

### 3.2. Result of the Literature Search

The review of Dayem et al., published in 2017, is the last published review that explains the effects of metal-based nanoparticles on ROS production. It mentioned that structural alterations due to metal-based NPs leads to different biological functions, specifically the generation of ROS, which is a main factor in metal-based-NP-induced toxicity, as well as the modulation of cellular signaling involved in cell death, proliferation, and differentiation (Figure 1 and Figure 4) [57]. However, the mechanism of bactericidal action of metal-based NPs has not been clearly determined. According to advances in research, metal-based NPs notably disrupt the synthesis of the cell wall and of proteins. In addition, NPs affect nucleic acids and cellular metabolic mechanisms. It was also hypothesized that the bactericidal effect of AgNPs may result from the induction of ROS production. It was described that the ions leached from AuNPs and AgNPs generate ROS, which create oxidative stress. The vital functionalities of the bacterial cell do not allow it to eliminate the excess oxidative stress created by superoxide radicals, hydroxyl radicals, and hydrogen peroxide. These radicals also change the permeability of the bacterial membrane (Figure 1). Furthermore, the ions released from these NPs also interact with the free amino, carboxyl and mercapto groups of proteins and nucleic acids. In addition, AuNPs and AgNPs also inhibit the electron transport chain of bacterial cells, which impairs ion exchange across the membrane and hence inhibition of metabolic activity. These processes enhance the death of bacterial cells and consequently the elimination of biofilms [57].

Zada et al., 2018, investigated the antibacterial activity of biosynthesized AuNPs (100–200 nm) against *E. coli* and *S. aureus* by well diffusion protocol. The results showed a zone of inhibition equal to 18 ± 2 mm and about 14 ± 2 mm against *E. coli* and *S. aureus*, respectively. The authors were interested in determining the mechanism through which AuNPs act; they evaluated generation of ROS. Bacterial cells exposed to AuNPs were treated with 10 mM of 2,7-dichlorodihydrofluorescein diacetate (DCFH-DA) dye. The intracellular generation of ROS was determined by fluorescence microscopy in the prepared samples. The samples were excited at 488 nm, which emits a green color at 535 nm in the presence of ROS. A significant amount of fluorescence was observed in bacterial cells exposed to AuNPs, but not for the control group. These observations suggested that AuNPs that reached the intracellular level stimulated the cell to generate ROS, which oxidized 2,7-dichlorofluorescein diacetate to a fluorescent material. The authors confirmed that metal-based NPs could be effective agents for generating ROS that induce cell death via the activation of an oxidative stress mechanism.

Further, membrane disruption was also explored to determine the effect of ROS generation by the biosynthesized AuNPs. This was tested using propidium iodide (PI) assay. PI can only penetrate cells with damaged membranes. To determine the existence of damaged bacterial cells, bacteria were treated with varying concentrations of AuNPs mixed with PI (30 μL/mL) for 30 min in the dark. After incubation, cells were placed on a microscope slide, excited at 535 nm, and red fluorescence at 639 nm was measured. The results revealed that bacterial cells treated with AuNPs exhibit intracellular fluorescence, indicating bacterial membrane damage and cell death [69].

Kadiyala et al., 2018, investigated the dose-response effect of ZnONPs on *S. aureus* cultured in planktonic conditions. The antibacterial activity was evaluated by ROS measurement. *S. aureus* cells were exposed to an increasing range of ZnONPs concentrations: 8, 80, 160, 400, and 800 μg/mL. ROS generation was quantified by the fluorescence of 3′-(p-aminophenyl) fluorescein (APF) and DCFH-DA probe. *S. aureus* exposure to ZnONPs at 800 μg/mL resulted in a greater than 1000× reduction in colony forming units (CFU) compared to the control (5 × 10^7^ CFUs of *S. aureus* without ZnONPs after 18 h of culture). The authors demonstrated a dose-dependent fluorescent response of the two ROS indicators in cultures exposed to NPs. The absolute fluorescence of APF (reflecting the amount of ROS produced) for low-dose ZnONPs at 80 μg/mL was slightly lower than that of pure water. In contrast, there was a strong fluorescence (marking a large fraction of APF bound to ZnONP) at high doses (800 μg/mL). The same results were determined using the DCFH-DA probe, although this probe appeared to be more reliable than ZnONPs as an ROS indicator. Indeed, when the concentration of NPs increased, the fluorescence induced by APF decreased, reflecting saturation of the dye [70].

Liao et al., 2019, investigated the antibacterial activity of AgNPs with diameters ranging from 5 to 20 nm on multidrug-resistant or drug-resistant *P. aeruginosa* clinical isolates. A bacterium was defined as multidrug resistant when resistant to three or more antibiotics. The MICs and MBCs of AgNPs at different concentrations (45, 22.5, 11.25, 5.625, 2.813, 1.406, 0.703, 0.352, 0.176, or 0.088 μg/mL) were determined against 22 strains of *P. aeruginosa* (reference strain American Type Culture Collection (ATCC) 27853 and 21 clinical isolates) with different bacterial inocula (10^3^, 10^5^, or 10^7^ CFU/mL). ROS generation was assessed at different time points (0, 0.25, 0.5, 0.75, 1, 1.5, and 2 h post AgNP treatment) on the *P. aeruginosa* strains (at 10^8^ CFU/mL) using DCFH-DA dye assay. Fluorescence was measured spectrophotometrically at λ = 520 nm. Results of the agar dilution test demonstrated that the mean MIC in the multidrug-resistant group was 2.285 ± 1.492 μg/mL, and the mean MBC was 3.165 ± 0.994 μg/mL. The mean MIC and MBC in the drug-resistant group was 2.596 ± 1.126 μg/mL and 3.246 ± 1.056 μg/mL, respectively. Experiments on different inocula of bacteria revealed that the MIC of AgNPs significantly rises with the increase of bacterial concentration; as well it was demonstrated that AgNPs could effectively kill the multidrug-resistant *P. aeruginosa* strains, and that effectiveness was positively correlated with the concentration of AgNPs. The labeling used to measure intracellular ROS production indicated that AgNPs induce elevated ROS production in multi-drug resistant *P. aeruginosa*, and this appears to be time- and concentration-dependent. Indeed, the fluorescence intensity of bacteria treated with AgNP increased with the extension of the contact time with AgNP throughout the 2 h. [71].

Nayak et al., 2019, synthesized AgNPs using medicinal plants indigenous to India: *Azadirachta indica* and *Syzygium cumini*. The antibacterial activity of these biosynthesized AgNPs (100–150 nm) was evaluated against *Bacillus subtilis* and *E. coli* by growth kinetics (OD600 nm) in the absence or presence of varying concentrations of the AgNPs (final concentration of 5, 10, 15, or 20% (*v*/*v*) of formulation in cultures). Specifically, AgNP formulations were added in the middle of the log phase of bacterial growth kinetics. Then, the ROS production in *E. coli* and *B. subtilis* upon treatment with different purified AgNPs (at the mid-log phase of the growth kinetics) was evaluated using a ROS-specific fluorescent DCFH-DA dye. The biosynthesized AgNPs concentration in growth medium significantly inhibited the growth of *B. subtilis* (log phase for >15 h) and *E. coli* (log phase for >8 h). In addition, an equivalent level of ROS was produced in both bacterial cultures upon treatment with varying concentrations of the AgNPs formulations. Hence, the increase of the NPs concentration added in the mid-log phase of growth promoted ROS generation by 25–50% [72].

Dong et al., 2019, synthesized AgNPs of different sizes (10 ± 5; 30 ± 5; 60 ± 5; 90 ± 5 nm) using eco-friendly reagents and studied their antibacterial properties against *Vibrio natriegens* by determining the MIC, bacterial membrane damage, ROS generation, and DNA damage.

To verify the antibacterial activity, the NP-treated bacterial cultures were plated on Muller–Hinton agar (MHA), and the MBCs were measured. The MBC is defined as the lowest concentration of NP that does not allow visible bacterial growth on agar plates at 37 °C after 24 h of incubation. For ROS detection, plates were incubated with nitroblue tetrazolium (NBT, 1 mg/mL) in Hanks balanced salt solution (HBSS) in the presence of AgNO_3_ solution or AgNPs at different concentrations and different sizes. Then, the solutions were incubated for 30 min at 37 °C in the dark, 0.1 M of HCl was added to stop the reaction, and the cells were treated with dimethylsulphoxide (DMSO) to extract reduced NBT. Finally, the solution was mixed with the salt solution, and the absorbance was measured at 575 nm to measure the production of intracellular ROS.

Data demonstrated that 10 ± 5 nm AgNPs can totally inhibit bacteria cell growth at a MIC equal to 1.0 μg/mL, while NPs with a size of about 90 ± 5 nm inhibit bacteria growth at a higher concentration (11.5 μg/mL). Among the different particle sizes, the MBC for 10 ± 5 nm AgNPs was the lowest (1.1 μg/mL) and that of 90 ± 5 nm Ag-NPs was the highest (11.7 μg/mL). These results indicated that the antibacterial properties of AgNPs were affected by size, with the smaller the particle size, the better the bactericidal effect. However, when the AgNPs size decreased, the OD 575 nm values increased, indicating that more ROS were produced by bacteria. These results strongly support the findings that AgNPs are valuable antibacterial agents and kill bacteria via ROS production [73].

Wang et al., 2019, tested the antibacterial activity of silicon nanowire arrays modified with Au/Ag NPs alloy (SN-Au/Ag) under photothermal effect against *E. coli* (MG1655). Data revealed that SN-Au/Ag with a mean diameter of 182.7 ± 9.6 nm killed bacteria during the 10 min sunlight exposure (intensity of the solar-simulating irradiation was 8.43 mW cm^2^); meanwhile, only 25.0 ± 3.4% and 25.5 ± 8.8% of the bacteria were killed when exposed separately to SN-Au and SN-Ag, respectively. The possible generation of ROS from the nanomaterials via light exposure was therefore studied. After exposure to sunlight for 10 min, SN-Au/Ag treated bacteria were mixed with DCFH-DA for 15 min at 37 °C. The ROS fluorescence intensity was determined at 488 nm (excitation) and 522 nm (emission). Results showed that exposure to sunlight for the same 10 min period increased the generation of hydrogen peroxide (H_2_O_2_) for all of the experiments. Experiments on the effect of H_2_O_2_ concentration on bacterial growth showed that the death rate was greater than 90% at concentrations higher than 0.5 mg/mL of H_2_O_2_. These data indicate that, in addition to the photothermal effect of the SN-Au/Ag nanomaterial, the photocatalytic generation of ROS also contributed to the death of bacteria exposed to sunlight [74].

In the work of Gunawan et al., 2020, the authors described the interactions of AgNPs (~2 nm) with bacterial cell wall components and ROS-mediated toxicity of these NPs. Generation of ROS and its implication in the alteration of the bacterial cell wall was revealed using electron paramagnetic resonance (EPR) spectroscopy in the AgNPs-only systems. This was performed by exposing AgNPs to isolated cell wall components: peptidoglycan (*B. subtilis*, code 69554), lipopolysaccharide (*E. coli* 0111:B4, code L2630), lipoteichoic acid (*B. subtilis*, code L3265), and phosphatidylethanolamine (*E. coli*). Each cell wall component (2 mg/L) was exposed to AgNPs at a concentration of 0.12 or 2.0 mg Ag/L. EPR allows the detection of oxygen radicals via the absorption of electromagnetic radiation emitted by the unpaired electrons present in the free radicals when they are stimulated by a magnetic field. The intensity of the signal is directly proportional to the concentration of ROS in the sample. ERP was performed first on the AgNPs system in the absence of cells in order to detect abiotically generated ROS. Next, it was performed on the cell-only system, with no added silver, for physiological cellular ROS detection. Finally, it was conducted on the AgNPs–cell system to determine nanoparticle-induced cellular ROS generation. All systems were treated with the spin probe for ROS EPR analysis. It appeared that the generation of abiotic ROS is most likely the mechanism of action by which AgNPs target cell wall components, resulting in disordered states of the fatty-acid tail of the phospholipid phosphatidylethanolamine (PE). AgNPs targeting of cell wall components, in particular the phospholipid, is most likely one of the key causes of the increased cellular ROS levels seen in many bacterial studies [89,90]. These membrane rearrangements are consistent with AgNP-induced inhibition of membrane-bound respiratory chain enzymes, which has been hypothesized to cause electron leakage into the cytoplasm and to reduce cytoplasmic molecular oxygen (O_2_) to superoxide radicals [91]. Exposure to AgNPs elevated generation of superoxide radicals in the model bacterium, with over 90% cell death detected in just 1 h of AgNPs exposure at the MIC. ROS is considered a factor that caused rapid killing in the bacterium model [75].

In the work of Karami et al., 2020, the antimicrobial properties of ZnO-based NPs (ZnO: 131.8 ± 9.7 nm; I: ZnO: 236.3 ± 40.6 nm; Ag: ZnO: 144.5 ± 23.5 nm; I: Ag: ZnO: 274.9 ± 20.1 nm; Ag: I: ZnO: 326.2 ± 50.4 nm) were evaluated against Gram-negative *E. coli* (MG1655) and Gram-positive *S. aureus* (USA300). The amount of photogenerated ROS from synthesized ZnO-based NPs was measured to uncover their antimicrobial mechanisms.

The antimicrobial efficacy of ZnO nanoparticles doped with I and Ag (Ag: I: ZnONPs) and AgZnO was determined by exposure to visible light and in the dark. Dark conditions were used as a control to evaluate the potential bacterial reduction due to the light source activating the ZnONPs. The quantity of photogenerated ROS after NP treatment was followed by 3′-p-(aminophenyl) fluorescein (APF), which spots hydroxyl radicals (·OH). APF is a non-fluorescent dye which exhibits a green fluorescence when reacting with ROS (e.g., ·OH, ONOO^-^…). Experiments showed that the Ag: ZnO, I: Ag: ZnO, and Ag: I: ZnO NP samples were very effective in killing *E. coli* and *S. aureus* when illuminated, with a CFU reduction ratio of 100%. On the contrary, ZnO and I: ZnO NP samples were less effective. In the dark, the Ag: ZnO NP sample was still very potent in eliminating *E. coli*. In contrast, ZnO, I: ZnO, I: Ag: ZnO, and Ag: I: ZnO NP samples had minor or moderate antimicrobial activity against both strains, *E. coli* and *S. aureus*. To determine the antimicrobial mechanisms of the ZnO-based NPs, results of the photo-generated ROS from ZnO-based NP samples found a significantly higher level of ROS generated from the Ag: I: ZnO and I: Ag: ZnO NP samples (fluorescence measured between 10^4^ and 10^5^) than the ZnO, I: ZnO (fluorescence measured between 10^3^ and 10^4^), and Ag: ZnO NP samples (no fluorescence) under light conditions. The results suggest that treatment of ZNO-based NP samples with iodine alone or silver alone, does not improve the photocatalytic activity of ZnO NPs. Moreover, the photocatalytic activity is significantly enhanced when the ZnO NPs are treated both with iodine and silver (i.e., Ag: I: ZnO NPs). The results in the dark confirmed a low level of ROS generation for all NP samples [76].

In the study of Wu et al., 2020, *Pseudomonas stutzeri* was used to explore the effects of AgNPs (15 ± 5 nm) on denitrification and cytotoxicity. *P. stutzeri* (ATCC 17588) was exposed to AgNPs at different doses (3.125, 6.25, or 12.5 mg/L) in denitrification medium. Lactate dehydrogenase (LDH) release and malondialdehyde (MDA) production were measured to determine cell membrane damage. LDH is used as marker of ROS generation, while MDA is used as a biomarker for lipid peroxidation (since it is generated by the degradation of polyunsaturated lipids). After treatment with AgNPs, intracellular ROS were detected with a DCFH-DA assay kit. The production of ROS in the cells was detected by the fluorescent signal (λ_ex_/λ_em_ = 485/535 nm).

The amount of Ag associated with membranes and cytoplasmic fractions of *P. stutzeri* were clearly increased to that of cellular membranes, suggesting that extracellular Ag could bind to the membrane very efficiently and then be internalized to produce intracellular ROS. Exposure to AgNPs significantly increased ROS production in *P. stutzeri* cells. This result showed that the toxicity of AgNPs to *P. stutzeri* was possibly related to the production of ROS. The possible mechanism of cytotoxicity is that AgNPs adhere to the surfaces of bacteria, resulting in membrane depolarization and lipid peroxidation. Furthermore, other authors have demonstrated that AgNPs can penetrate the cell membrane and enter the cell, resulting in the generation of intracellular ROS [77].

Liu et al., 2020, studied the synergistic toxicity of ball-milled biochar (BMB) with CuONPs on *Streptomyces coelicolor* M145. They explored how the merged properties of CuONPs and BMB at different pyrolysis temperatures affected cell viability, oxidative stress, membrane damage, antibiotic production, and gene expression. This study provides insights into the synergistic toxicity of mixtures of BMB and metal-based NPs. *S. coelicolor* M145 was prepared and treated with specified concentrations of nanomaterials. Considering the adsorption capacity of BMB, the concentrations of CuONPs were set at 10 and 20 mg/L. DCFH-DA was used as the indicator of intracellular ROS production. Fluorescence intensities were measured through excitation/emission wavelengths of 488/525 nm.

Results showed that each treatment significantly increased the ROS level compared with that of the control. In detail, after cotreatment with CuONPs the ROS level increased based on different types of BMB (300 °C, 500 °C, and 700 °C); with the increase of 40 nm CuONP concentration (10 and 20 mg/L) ROS levels further increased. In the end, all kinds of BMB significantly increased the ROS level. In the same way, there was a significant dose–response relationship between the exposed materials and permeability of bacteria M145 [78].

Lv et al., 2020, studied the antibacterial activity of CuONPs doped with Mg, Zn, and Ce ions sized 100, 150, and 50 nm, respectively, against *S. aureus* and *E. coli*. The improved antibacterial activity of doped CuONPs was attributed to the synergistic effect of ROS generation and inactivation of protein in bacterial cells by the binding of Cu^2+^ ions to the bacterial cell surface.

In order to identify the antibacterial mechanism of action of the doped CuONPs, protein concentration and catalase activity in *E. coli* and *S. aureus* were determined. The catalase activity of the two models of bacteria was significantly increased during the exposure of the latter to doped CuONPs. The data revealed an increase in H_2_O_2_ in the treated cells as compared to the control. This gives evidence that the doped CuONPs were able to stimulate the generation of intracellular ROS. Moreover, the decreased protein concentration in *E. coli* and *S. aureus* as compared to the control showed that doped CuONPs effectively inactivated proteins in the cells, indicating that ROS increased, in turn enhancing the oxidative stress [79].

Mazur et al., 2020, studied the synergistic antibacterial activity of gentamicin and Tween-stabilized AgNPs against gentamicin-resistant clinical strains of *Staphylococcus epidermidis* (MDR, clinical strains), and then the production of ROS was measured. The generation of ROS was investigated by mixing luminol solution with Tween-stabilized AgNPs and gentamicin water solution (40 mg/mL) to obtain a final antibiotic concentration of 0.4 mg/mL. Following treatment by AgNPs and gentamicin, Tween-stabilized AgNPs were added to a luminol-based detection system to stimulate ROS production. The consecutive addition of gentamicin significantly increased the generation of ROS. Results showed that generation of ROS by Tween-coated metallic AgNPs was significantly enhanced by gentamicin [54].

Li et al., 2020, obtained *N,N*,*N*-trimethylammonium bromide-gold nanoclusters (MUTAB-AuNCs). MUTAB-AuNCs possess highly positive surface charges and near-infrared (NIR) luminescence properties. The authors found that these MUTAB-AuNCs exhibit a broad-spectrum antimicrobial effect without any detectable toxicity to mammalian cells.

Antibacterial activity of AuNCs was tested against several representative kinds of bacteria and fungi. *B. subtilis* (ATCC 6633), *E. faecalis* (ATCC 29212), *S. pneumoniae*, vancomycin-resistant *Enterococcus* (VRE), *P. aeruginosa*, *E. coli* (BL21), and *M. albican* were evaluated on agar plates by determining MICs. The MIC was defined as the lowest concentration of AuNCs that completely inhibited the visible growth of microbes and did not show a difference between the tested strains. MUTAB-AuNCs were effective against both Gram-positive and Gram-negative bacteria (*S. pneumoniae*, *P. aeruginosa*, and *E. coli*) and fungi (*M. albican*) with an MIC no larger than 4 mg/mL. Different treatments were used to determine the antimicrobial effects of MUTAB-AuNCs, such as a biofilm test, DNA leakage test, and fluorescence. The authors were interested in investigating ROS accumulation during treatment of *E. coli* with the complex MUTAB-AuNCs. For that, the bacterial ROS level was measured by an ROS assay kit based on DCFH-DA. Bacteria were incubated with DCFH-DA, and then MUTAB-AuNCs were added. The fluorescence of the solution was measured by spectrofluorometry. An increase of intracellular ROS was obtained compared with the control group. The authors suggest that the internalization of MUTAB-AuNCs can disturb cellular metabolism of bacteria by influencing multiple genes, such as genes related to the oxidation process and cell death [28].

Muñoz-Villagrán et al., 2020, aimed to design Au nanostructures (AuNS) with antimicrobial properties. For that, *E. coli* (BW25113, SP11, and GS022) was grown aerobically or anaerobically and exposed to chloroauric acid (HAuCl_4_) (250, 125, or 62.5 µM) or to potassium tellurite (K_2_TeO_3_) (2 µM). After the treatment protocol, the fluorescence intensity related to oxidation-sensitive probes DCFH-DA and dihydroethidine (DHE) was determined using excitation and emission wavelengths of 490 and 527 nm, respectively. Aerobically and anaerobically generated ROS were assessed using the oxidation-sensitive probes DCFH-DA and DHE, which allowed for determination of total ROS and superoxide ions, respectively. In aerobic conditions, the Au^3+^ MIC was 250 μM and the growth inhibition area was 1.12 ± 0.02 cm^2^. Minor changes to cell growth were observed when *E. coli* was exposed to lower Au^3+^ concentrations (7.8–15.6 µM). In addition, significant ROS formation was demonstrated in *E. coli* exposed to HAuCl_4_ under both aerobic and anaerobic conditions; hence, the increase in superoxide levels was dependent on the increase in Au^3+^ concentrations [80].

Da Silva et al., 2020, studied the interaction of metal-based NPs with light inducing local-surface plasmon resonance. LSPR excitation with visible-light emitting diodes (LED) enhanced the antibacterial effect of citrate-covered AgNPs against *Pseudomonas aeruginosa*. Data revealed that the synthesized AgNPs, with a diameter of 20 ± 3 nm, displayed strong antimicrobial activity against *P. aeruginosa*, and that the MIC of the AgNPs was 10 μg/mL. When the AgNPs were exposed to light, the MIC dropped to 5 μg/mL. The MBC was 20 μg/mL and 10 μg/mL in dark and light conditions, respectively. To test whether AgNPs caused oxidative stress, the authors evaluated the intracellular level of ROS in *P. aeruginosa*. Bacteria were treated with 10 μg/mL AgNPs in light or dark conditions, followed by exposure to DCFH-DA. After incubation at 30 °C for 1 h in either the presence or absence of light, the fluorescence emitted by the intracellular oxidation of the dye was determined. The level of ROS increased by 2.5-fold (dark) and 4.8-fold (light) when compared to the respective controls. According to this experiment, it seems that AgNPs, which have antibacterial effects against *P. aeruginosa*, were enhanced by the formation of ROS via LSPR. Moreover, the release of Ag^+^ occurred several hours after AgNPs had already killed the bacteria [81].

Yang et al., 2020, were interested in testing the antibacterial and anti-osteosarcoma effects of zinc oxide-3-carboxyphenylboronic acid-naringin (ZnOPBA-NG) NPs in the coculture model of bacteria and osteosarcoma cells. *S. aureus* (ATCC 29213) and *E. coli* (ATCC 25922) were used to determine antibacterial activity, and DCFH-DA (10 μM) was added to determine the production of ROS in the coculture in the presence of a Ti substrate (Ti-ZnO-PBA-NG). The absorbance of the mixture was measured with a UV–vis spectrophotometer at λ = 260 nm.

Results showed that Ti-ZnO-PBA-NG displayed the highest antibacterial activity. The relevant antibacterial capacities were 94.1% and 95.5% against *E. coli* and *S. aureus*, respectively. The *E. coli* and *S. aureus* biofilms on the Ti surface was evaluated with crystal violet dye staining. Quantitative analysis with UV–vis spectroscopy showed *E. coli* and *S. aureus* biofilm on all surfaces. Blue color decreased remarkably on Ti-ZnO, Ti-ZnO-PBA, and Ti-ZnO-PBA-NG surfaces. The biofilms formed on all modified Ti substrates were significantly reduced, indicating that functionalizing Ti substrates with ZnO-PBA-NG nanoparticles could more effectively inhibit the formation of bacterial biofilms. In addition, the results showed that, compared to native Ti, the Ti substrate modified with ZnO nanoparticles induced the production of a significantly higher level of intercellular ROS. The accumulation of ROS in bacteria and osteosarcoma cells resulted in damage of nucleic acids, amino acids, and membrane lipids. Naringin (NG) and Zn^2+^ induced a remarkable increase in oxidative stress in bacteria and osteosarcoma cells by producing ROS. Accumulation of ROS resulted in damage of bacteria and induced apoptosis in the osteosarcoma cells [82].

Perveen et al., 2021, evaluated levels of ROS generated in *Listeria monocytogenes* (ATCC 19114) and *Serratia marcescens* (ATCC 13880) upon treatment with AuNPs. Biofilm inhibition was evaluated using crystal violet assay staining at 0.006–0.5 × MICs. ROS generation was measured by DCHF-DA probe. Firstly, MICs were evaluated against the two bacteria using AuNPs with an average size of 24.4 nm fabricated from the seed extract of *Trachyspermum ammi* (TA-AuNPs). Results demonstrated that the synthesized TA-AuNPs exhibited MICs of 16 and 32 µg/mL against *S. marcescens* and *L. monocytogenes*, respectively. Secondly, the biofilm inhibition test showed a reduction in biofilm formation of 81% and 73% (recorded at 0.5 × MIC) for *S. marcescens* and *L. monocytogenes*, respectively. Intracellular ROS generated in TA-AuNP-treated cells increased by 59% and 51% in *L. monocytogenes* and *S. marcescens*, respectively. The authors suggested that increased ROS production could be one of the most important mechanisms through which AuNPs disrupt the normal functioning of bacterial cells [83].

Samuggam et al., 2021 demonstrated the antibacterial activity of plant-based AgNPs against MDR bacteria. The synthesized AgNPs were characterized and further tested for their antibacterial and ROS properties. Characterization showed the AgNPs to be between 8 to 50 nm. Antibacterial activity was evaluated using a Kirby–Bauer antibiotic test (disc diffusion test) against selected Gram-positive (*S. haemolyticus*, *S. epidermidis*, *B. subtilis*, *S. aureus*, and *S. pyogenes*) and Gram-negative bacterial strains (*P. mirabilis*, *V. cholerae*, *K. pneumoniae*, *E. coli*, *P. aeruginosa*, *E. cloacae*, and *S. typhi*). ROS production was analyzed using the following protocol: a total of 200 mL of each bacterial strain was treated with AgNPs and kept at 37 °C. After this step, the bacterial suspension was centrifuged to obtain the pellet, which was treated with a 2% nitro blue tetrazolium (NBT) mixture and kept at room temperature for 60 min in dark conditions. The cell-membrane pellet was disrupted by treatment with 2 M KOH solution. The absorbance was measured at 620 nm. Results demonstrated that AgNPs synthesized from *Sondias mombin* leaf extract showed high antimicrobial activity only against *S. epidermidis* (23.65 ± 0.35 mm), *Salmonella typhi* (23.67 ± 0.33), *P. mirabilis*, *Enterobacter cloacae*, *E. coli*, and *P. aeruginosa* (21 mm). Based on ROS quantification, AgNPs showed a significant ROS level in *S. typhi* and *V. cholerae* [84].

Srinivas Naik el al., 2021, studied the antimicrobial activity of AgNPs with an average size of 13.2 nm biosynthesized using the root extract of the perennial plant “Spiny gourd” (*Momordica dioica*). The antibacterial study of AgNPs was performed against: *B. subtilis* (ATCC 6051), *S. aureus* (ATCC 29736), *P. aeruginosa* (ATCC 27858), *E. coli* (ATCC 8739), *K. planticola* (ATCC 2719), and *C. albicans* (ATCC 2091). The MIC assay of the biosynthesized AgNPs was performed by the broth microdilution method, and intracellular ROS accumulation was measured using DCFH-DA probe. The concentration of the bacteria treated with AgNPs was adjusted to 10^5^ CFU/mL and incubated at 37 °C for 4 and 8 h. After centrifugation, a volume of 100 μM of DCFH-DA was added to each supernatant, which was then incubated for 1 h. The concentration of ROS accumulated in the supernatant was measured using spectrofluorimetry.

The AgNPs demonstrated a significant antibacterial activity against *S. aureus* and *E. coli*, including at the lowest concentrations tested (i.e., 3.125 µg/mL and 6.25 µg/mL) compared to control. Moreover, the antimicrobial activity of the nanoparticles increased as their size decreased. The authors suggested that this may be due to the increase in surface area per unit volume. In addition, the biosynthesized AgNPs also influenced ROS accumulation, which induced membrane damage in the pathogens. This is reflected by the fact that the concentration of ROS in both pathogens increased 0.9-fold compared to the control sample (i.e., pathogens not treated with AgNPs) [85].

Li et al., 2021, developed novel 200 nm, ROS-generating, fluorescent, nanoparticle-based, CuInS/ZnS quantum dots. The bactericidal activity of this compound was investigated against four strains of *P. aeruginosa*: PA14, PAO1, PAK, and one MDR strain. After incubation with CuInS/ZnS for 3 h, bacteria were found under 660 nm illumination to be labeled with red fluorescence, which indicated that the nanoparticle-based CuInS/ZnS influenced bacterial adhesion, causing bacterial aggregation. In order to evaluate the ROS-generating ability of CuInS/ZnS, the non-fluorescent DCFH-DA was examined under 660 nm irradiation (1 W/cm^2^). In CuInS/ZnS-treated bacteria, results showed much stronger fluorescence intensity than the control group (untreated bacteria). As the irradiation time was extended, the nanobiocide-based CuInS/ZnS exhibited a gradual increase in fluorescence intensity, which demonstrated the accumulation of ROS and alteration of glycoproteins and lipids on the cytomembrane of pathogenic bacteria [86].

Metryka et al., 2021, studied the antibacterial activity of commercial (obtained from Sigma-Aldrich company, St. Louis, MO, USA) AgNPs (cat. 576832), CuNPs (cat. 774081), ZnONPs (cat. 677450), and TiO_2_NPs (cat. US1019F) with sizes of <100 nm, 25 nm, <50 nm, and 20 nm, respectively, against *E. coli* (ATCC 25922), *B. cereus* (ATCC 11778), and *S. epidermidis* (ATCC 12228). Before the treatment of the bacteria, dilutions of the NPs were prepared in sterile LB medium, with final concentrations from 0 to 1200 mg/L. Then, increasing concentrations of the commercial NPs were added to each bacterial culture. Inhibition of bacterial growth was established using a mortality rate formula based on MIC and MBC values. Moreover, the total concentration of ROS and levels of singlet oxygen (^1^O_2_), superoxide radical anion (O_2_·^−^), hydrogen peroxide (H_2_O_2_), and hydroxyl radical (·OH) were measured in samples treated with AgNPs, CuNPs, ZnONPs, and TiO_2_NPs. In addition, the total intracellular ROS concentration was evaluated using DCFH-DA dye assay.

Concerning antibacterial activity, Gram-positive strains reacted similarly to AgNPs (*E. coli*: 10 mg/L; *B. cereus*: 850 mg/L; *S. epidermidis*: 500 mg/L), CuNPs (*E. coli*: 200 mg/L; *B. cereus*: 75 mg/L; *S. epidermidis*: 200 mg/L), ZnONPs (*E. coli*: 425 mg/L; *B. cereus*: 800 mg/L; *S. epidermidis*: 750 mg/L), and TiO_2_NPs (*E. coli*: 500 mg/L; *B. cereus*: 100 mg/L; *S. epidermidis*: 1050 mg/L). In addition, the production of ROS induced by treatment of *E. coli*, *B. cereus*, and *S. epidermidis*, with AgNPs, CuNPs, ZnONPs, and TiO_2_NPs, was measured. In the end, it appears that the results are very different depending on the bacteria used, the nanoparticle tested and the ROS produced. Overall, Metryka and colleagues demonstrate that most of these NPs induce a strong production of all ROS (i.e., H_2_O_2_, ·OH, ^1^O_2_, O_2_·^−^) mainly in *S. epidermidis*; while this increase is more contrasted in *B. cereus* and *E. coli*. Conversely, CuNPs seem to be the particles most likely to stimulate the production of ROS, regardless of the bacteria tested [87].

Recently, Meng et al., 2021, studied bacterial redox balance under treatment by AU noble metal nanoclusters (NC) of an average size of 1.5 nm with cinnamaldehyde (CA) on the surface. Two *S. aureus* strains were used, Methicillin-susceptible *S. *aureus** (MSSA, CMCC(B) 26003) and methicillin-resistant *S. aureus* (MRSA, ATCC 43300). A volume of 900 µL of the bacterial suspension was mixed with 100 µL of the Au NC/CA-Au NC solution (with a final concentration of Au within 10–80 µM). The authors tested bacterial membrane damage using Hoechst 33342 (5 µg/mL) and propidium iodide (PI) (10 µg/mL) on MSSA bacteria which had been exposed to 30 µM Au NC/CA-AuNC solutions for 15 min in the dark. In addition, the intracellular ROS levels were evaluated using DCFH-DA dye. The fluorescence intensity was measured at wavelengths of 488/525 nm. Experimentation revealed that CA-AuNCs exhibit an antibacterial effect through multiple antibacterial mechanisms, including disruption of membrane integrity, dissipation of membrane potential, and increased membrane permeability. Interestingly, the CA-AuNCs simultaneously enhanced ROS generation [88].

## 4. Implications of ROS in Microbial Therapeutic Clinical Trials

Growing evidence suggests that ROS production could be a novel therapeutic strategy to combat microbes; with the long-term hope of decreasing the volume of antibiotics used in medicine, agriculture, and the environment (Figure 1). In several studies, the authors demonstrated that ROS are highly antimicrobial against both Gram-positive and Gram-negative bacteria. Much clinical investigation remains to be carried out on ROS therapy (Figure 4) [92]. Indeed, even if clinical trials on the antibacterial properties of NPs, admittedly in limited number, exist, to date none has demonstrated the effectiveness of oxidative stress induced by metal-based NPs as part of a treatment for bacterial infection (i.e., no results were found on the clinical trials database, https://clinicaltrials.gov/ (accessed on 1 February 2022) using keywords of “infection”, “bacterial”, “fungi”, “nanoparticle”, “metal”, or “ROS”.

The first approved ROS therapeutic agent was in the form of a pharmaceutical honey wound gel, Surgihoney Reactive Oxygen (SurgihoneyRO™, Melton, UK). This product has been engineered to provide a constant level of ROS over a prolonged period of time when applied to a wound [93]. Moreover, ROS used in inhaled form have multiple targets for therapeutic use, primarily to reduce biofilm in chronic respiratory diseases, such as chronic bronchitis, cystic fibrosis, bronchiectasis, and ventilator-associated pneumonia [94]. In addition, therapy based on ROS production is used in surgical prophylaxis and prosthetic joint infections. A treatment targeting ROS production has been developed. Photodynamic inactivation (PDI) was used to cure leg ulcers. PDI led to ROS production, which was responsible for the death of microorganisms, including those present in the form of biofilm in chronic wounds. It also decreased the leg ulcers, and thus remarkably improved the health of the patient. [95]. Based on their use in different clinical trials, ROS have emerged as one of the few antibiotic alternatives to reach clinical use. Physiologic ROS production is involved in several essential signaling functions targeting prokaryote and eukaryote cells. ROS remain a key signaling molecules in redox homeostasis, which regulates cell growth, cell death (necrosis and apoptosis), survival, and mitochondrial metabolism [96]. Mitochondria-derived ROS are involved in the activation of AMP protein kinases, which are implicated in several cancer signaling pathways [97]. However, ROS are among several other intracellular signaling transducers that sustain and control cell physiology. The elucidation of diseases’ causal molecules and activated signaling pathways will lead to the development of combination therapies based on mechanisms targeting the activation of ROS [98] (Figure 1 and Figure 4).

## 5. Conclusions

Considerable progress has been made in the last few years for the use of ROS in conjunction with metal-based nanoparticles as bactericidals. There is still no well-developed technology to assess the ability of ROS to kill bacteria during treatment with metal-based NPs, and the associated therapeutic effects. The newest strategy is to develop ROS-generating, nanoparticle photosensitizers for targeting MDR bacteria, but a precise ROS therapy system still needs to be defined [86]. As has been reported, the most common method to assess ROS production is still dichlorofluorescein and its esterified analogue (DCFH-DA) (Table 1). This probe relies on two cellular catalytic activities that are prerequisite for its reactivity with hydrogen peroxide. Some clinical studies also demonstrated the bactericidal effects of ROS in clinical trials of wounds and in the reduction of biofilm during respiratory infections, but the story is far from over [92,94,95,96]. It will be interesting to develop future ROS-based strategies as clinical therapies for microbial infections.

## Figures and Tables

**Figure 1 microorganisms-10-00437-f001:**
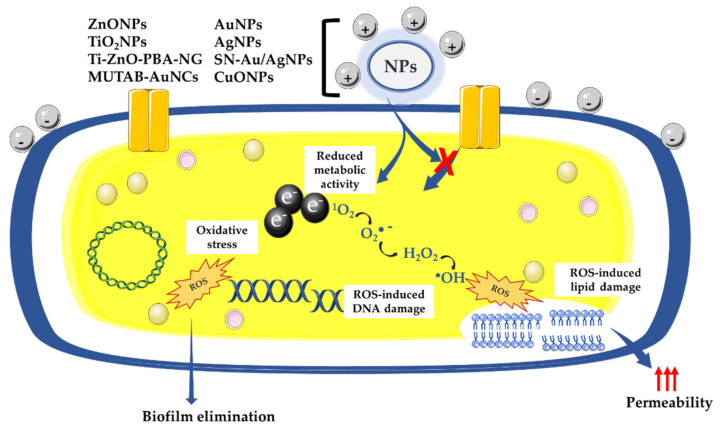
Antibacterial effect of oxidative stress mediated by metal-based NPs: NPs, nanoparticles; AuNPs, gold nanoparticles; AgNPs, silver nanoparticles; SN-Au/AgNPs, silicon nanowire arrays modified with Au/Ag alloy NPs; CuNPs, copper nanoparticles; ZnONPs, zinc oxide nanoparticles; TiO_2_NPs, titanium dioxide nanoparticles; Ti-ZnO-PBA-NG, zinc oxide-3-carboxyphenylboronic acid-naringin; MUTAB-AuNCs, *N*,*N*,*N*-trimethylammonium bromide-gold nanoclusters; ROS, reactive oxygen species; ^1^O_2_, singlet oxygen.

**Figure 2 microorganisms-10-00437-f002:**
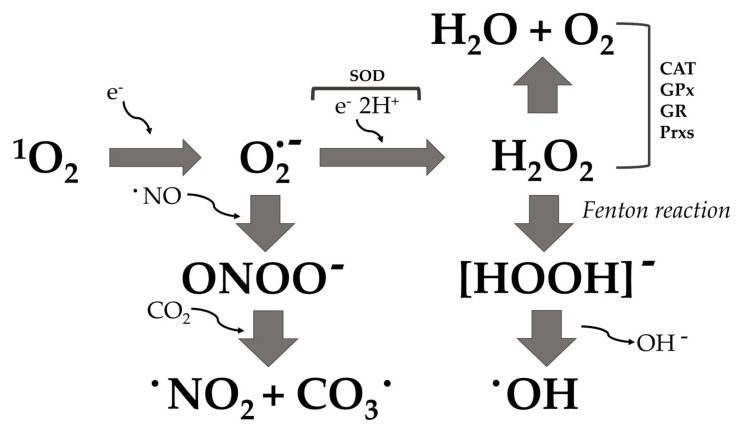
ROS production pathways: ^1^O_2_, singlet oxygen; O_2_·^−^, superoxide anion; ·OH, hydroxyl radical; ONOO−, peroxynitrite anion; ·NO_2_, nitrogen dioxide radical; CO_3_·, carbon trioxide. SOD, superoxide dismutase; CAT, catalase; GPx, glutathione peroxidase; GR, glutathione reductase; Prxs, peroxiredoxins.

**Figure 3 microorganisms-10-00437-f003:**
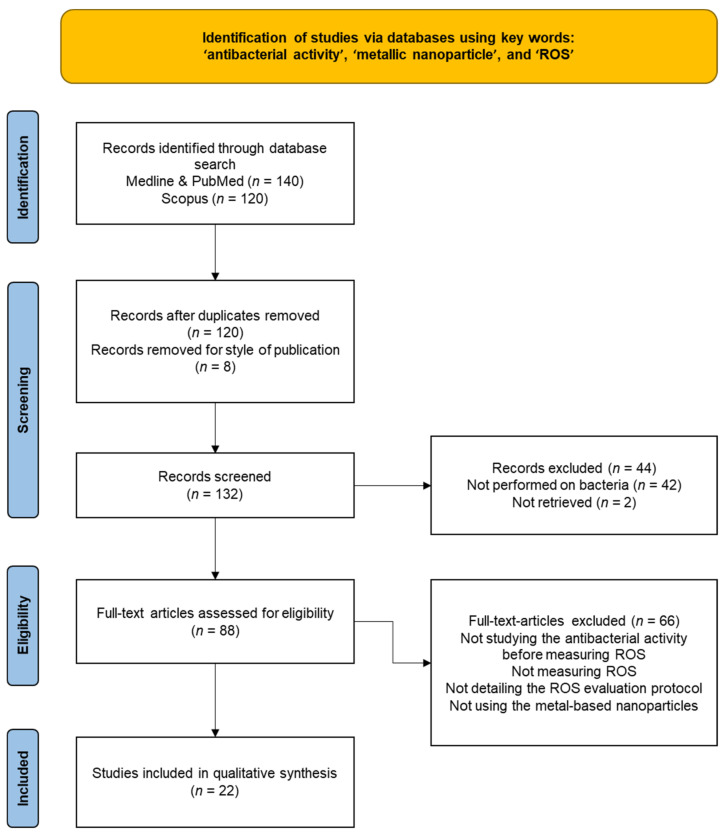
PRISMA flow diagram details of manuscripts retrieved via searches, abstracts screened, full-text articles assessed and used.

**Figure 4 microorganisms-10-00437-f004:**
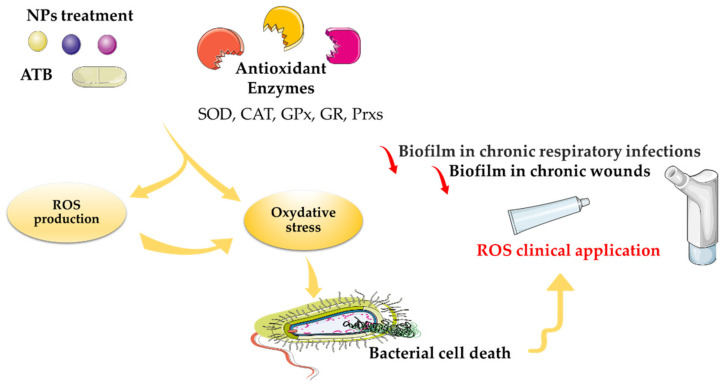
Involvement of ROS in the alteration of cell physiology and its clinical implications.

**Table 1 microorganisms-10-00437-t001:** Summary of metal-based nanoparticles inducing ROS production in bacteria.

o	Size of Nanoparticles	Bacteria Used	ROS Measurement	Reference
AuNPs	100–200 nm	*E. coli* and *S. aureus* ^#^	DCFH-DA dye assay	[69]
ZnONPs	-	*S. aureus*, substrain COL	APF and DCFH-DA dye assay	[70]
AgNPs	5–20 nm	*P. aeruginosa* (21 clinical isolates)	DCFH-DA dye assay	[71]
Biosynthesized AgNPs (*Azadirachta indica* and *Syzygium cumini*)	100–150 nm	*B. subtilis* (MTCC 736) and *E. coli* (MTCC 443)	DCFH-DA dye assay	[72]
AgNPs	10 ± 5 nm; 30 ± 5 nm; 60 ± 5 nm	*V. natriegens*	HBSS and NBT assay	[73]
SN-Au/Ag (under photothermal effect)	182.7 ± 9.6 nm	*E. coli* (MG1655)	DCFH-DA dye assay	[74]
AgNPs	~2 nm	*B. subtilis* 69554, *B. subtilis* L3265, *E. coli* 0111:*B4*_L2630	EPR spectroscopy	[75]
ZnONPs	131.8 ± 9.7 nm	*E. coli* MG1655 and *S. aureus* USA300	APF probe assay	[76]
I: ZnONPs	236.3 ± 40.6 nm	*E. coli* MG1655 and *S. aureus* USA300	APF probe assay	[76]
Ag: ZnONPs	144.5 ± 23.5 nm	*E. coli* MG1655 and *S. aureus* USA300	APF probe assay	[76]
I: Ag: ZnONPs	274.9 ± 20.1 nm	*E. coli* MG1655 and *S. aureus* USA300	APF probe assay	[76]
Ag: I: ZnONPs	326.2 ± 50.4 nm	*E. coli* MG1655 and *S. aureus* USA300	APF probe assay	[76]
AgNPs	15 ± 5 nm	*P. stutzeri* (ATCC 17588)	DCFH-DA dye assay	[77]
BMB-CuONPs	-	*Streptomyces coelicolor* M145	DCFH-DA dye assay	[78]
Mg-doped CuONPs	100 nm	*E. coli* and *S. aureus*	Protein concentration and catalase activity	[79]
Zn-doped CuONPs	150 nm	*E. coli* and *S. aureus*	Protein concentration and catalase activity	[79]
Ce-doped CuONPs	50 nm	*E. coli* and *S. aureus*	Protein concentration and catalase activity	[79]
Tween-stabilized AgNPs	20–40 nm	*S. epidermidis* (9 clinical isolates)	Chemiluminescence of luminol assay	[54]
MUTAB-AuNCs	-	*B. subtilis* (ATCC 6633), *E. faecalis* (ATCC 29212), *S. pneumoniae* (clinical isolate), *VRE* (clinical isolate), *P. aeruginosa* (clinical isolate), *E. coli* (BL21) and *M. albican*	DCFH-DA dye assay	[28]
AuNS	-	*E. coli* (BW25113, SP11, and GS022)	DCFH-DA dye assay	[80]
Citrate-covered AgNPs (light excitable)	20 ± 3 nm	*P. aeruginosa* (strain PA14)	DCFH-DA dye assay	[81]
Ti-ZnO-PBA-NG	-	*E. coli* (ATCC 25922) and *S. aureus* (ATCC 29213)	DCFH-DA dye assay	[82]
TA-AuNPs	24.4 nm	*Listeria monocytogenes* (ATCC 19114) and *Serratia marcescens* (ATCC 13880)	DCFH-DA dye assay	[83]
AgNPs (*S. mombin* leaf extract)	8–50 nm	*S. haemolyticus*, *S. epidermidis*, *S. aureus*, *S. pyogenes*, *B. subtilis*, *P. mirabilis*, *V. cholerae*, *K. pneumoniae*, *E. coli*, *P. aeruginosa*, *E. cloacae*, *S. typhi*	NBT assay	[84]
AgNPs	13.2 nm	*B. subtilis* (ATCC 6051), *S. aureus* (ATCC 29736), *P. aeruginosa* (ATCC 27858), *E. coli* (ATCC 8739), *Klebsiella planticola* (ATCC 2719), and *C. albicans* (ATCC 2091)	DCFH-DA dye assay	[85]
Nanobiocide-based CuInS/ZnS quantum dot	200 nm	*P. aeruginosa*: PA14, PAO1, PAK and one MDR strain	DCFH-DA dye assay	[86]
Commercial AgNPs (cat. 576832) (Sigma-Aldrich company, St. Louis, MO, USA)	<100 nm	*E. coli* (ATCC 25922), *B. cereus* (ATCC 11778) and *S. epidermidis* strains (ATCC 12228)	DCFH-DA dye assay singlet oxygen (^1^O_2_), superoxide radical anion (O_2_·^−^), hydrogen peroxide (H_2_O_2_), and hydroxyl radical (·OH) measurement	[87]
Commercial CuNPs (cat. 774081) (Sigma-Aldrich company, St. Louis, MO, USA)	25 nm	*E. coli* (ATCC 25922), *B. cereus* (ATCC 11778) and *S. epidermidis* strains (ATCC 12228)	DCFH-DA dye assay singlet oxygen (^1^O_2_), superoxide radical anion (O_2_·^−^), hydrogen peroxide (H_2_O_2_), and hydroxyl radical (·OH) measurement	[87]
Commercial ZnONPs (cat. 677450)	<50 nm	*E. coli* (ATCC 25922), *B. cereus* (ATCC 11778) and *S. epidermidis* strains (ATCC 12228)	DCFH-DA dye assay singlet oxygen (^1^O_2_), superoxide radical anion (O_2_·^−^), hydrogen peroxide (H_2_O_2_), and hydroxyl radical (·OH) measurement	[87]
Commercial TiO_2_NPs (cat. US1019F) (Sigma-Aldrich company, St. Louis, MO, USA)	20 nm	*E. coli* (ATCC 25922), *B. cereus* (ATCC 11778) and *S. epidermidis* strains (ATCC 12228)	DCFH-DA dye assay singlet oxygen (^1^O_2_), superoxide radical anion (O_2_·^−^), hydrogen peroxide (H_2_O_2_), and hydroxyl radical (·OH) measurement	[87]
AuNCs/CA-AuNC	1.5 nm	*S. aureus* (MSSA, CMCC(B) 26003) and (MRSA, ATCC 43300)	DCFH-DA dye assay	[88]

^#^ in many studies, the origin of the bacterial strains is not mentioned (e.g., strains from collections: ATCC ... or clinical isolates ...); we have provided the information where possible: NPs, nanoparticles; AuNPs, gold nanoparticles; AgNPs, silver nanoparticles; SN-Au/Ag NPs, silicon nanowire arrays modified with Au/Ag alloy NPs; CuNPs, copper nanoparticles; ZnONPs, zinc oxide nanoparticles; I, iodine; TiO_2_NPs, titanium dioxide nanoparticles; Ti-Zn0-PBA-NG, zinc oxide-3-carboxyphenylboronic acid-naringin; MUTAB-AuNCs, *N,N*,*N*-trimethylammonium bromide-gold nanoclusters; BMB-CuONPs, ball-milled biochar-copper oxide nanoparticles; Mg, magnesium; Zn, zinc; Ce, cerium; TA, *Trachyspermum ammi*; CA, cinnamaldehyde; ^1^O_2_, singlet oxygen; O_2_·^−^, superoxide anion; ·OH, hydroxyl radical; DCFH-DA, 2,7-dichlorodihydrofluorescein diacetate; APF, 3′-(p-aminophenyl) fluorescein; HBSS, Hanks balanced salt solution; NBT, nitroblue tetrazolium; EPR, electron paramagnetic resonance. MTCC, Microbial Type Culture Collection and Gene Bank, India. ATCC, American Type Culture Collection; VRE, vancomycin-resistant *Enterococci*; MSSA, methicillin-susceptible *Staphylococcus aureus*; MRSA, methicillin-resistant *Staphylococcus aureus.*

## Data Availability

Not applicable.

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
