# Peer review of "Current Knowledge on the Oxidative-Stress-Mediated Antimicrobial Properties of Metal-Based Nanoparticles"

_microorganisms, 2022, doi:10.3390/microorganisms10020437_

Round 1

Reviewer 1 Report

This research is under the scope of this journal; the topic is relevant for readers, and this research deals with potentially significant knowledge of the field. 

However, there are some concerns about the present manuscript: 

(Introduction)

What is the importance of this review study? What is the gap in this field of literature?

    • You do not think this study is included in the others already done? Which results are comparable with other studies? What has this study been new?
  • Improve the resolution quality of all figures and graphs (and a presentation). The font/ language in the figure/caption is different from the text.

  • Many Antimicrobial peptides (AMPs), such as LL37 peptides, may be immobilized nanoparticles (AuNPs) on the surface of medical devices, to render them with antimicrobial and angiogenic properties.  Please read this https://doi.org/10.1039/D1BM01034D and https://doi.org/10.1039/C0JM00817F 

(M&M) 

  • You need to explain the process of articles selection, using a PRISMA flow diagram.  We need to know what was the reasons for exclusions or inclusion. 

(Results) 

- on Table 1: add a column with Authors and year. 

(References)

  • Need to add more references in the manuscript, see the articles recommend.
  • Check reference’s format MDPI in the manuscript, and in the references. The titles of references have a different format, the title of the article is written in capital letters at the beginning of words, others only in lower case. Also, the standardized format of presentation in the journal's name. Because names have been written in a different format, one is not abbreviated, others are not.

Author Response

Dear Reviewer 1

Thanks

Reviewer 2 Report

The manuscript is overall well written with current informations. The author has provided an update on the state of the art on research dealing with the study and characterization of the induction of oxidative 23 stress-mediated antimicrobial mechanisms by metal-based NPs. This current review article will provide latest information on nanoparticle based antimicrobial properties with emphasis on the ROS. The author need to improve the figure quality and obtain copyright permission from the original source if adopted from previously published literature.

Author Response

Dear Reviewer 2

Thanks

Round 2

Reviewer 1 Report

This research is under the scope of this journal; the topic is interesting for readers.

The authors improved the quality of the manuscript after the reviewer's indications.